# Understanding parents' decision-making on participation in clinical trials in children's heart surgery: a qualitative study

Nigel E Drury [ID],[1,2] Julie C Menzies [ID],[3] Clare J Taylor [ID],[4] Timothy J Jones [ID],[1,2] Anna C Lavis [ID][5]

NED and JCM contributed equally.

[1]Paediatric Cardiac Surgery, Birmingham Children's Hospital, Birmingham, UK
[2]Institute of Cardiovascular Sciences, University of Birmingham, Birmingham, UK
[3]Paediatric Intensive Care, Birmingham Children's Hospital, Birmingham, UK
[4]Nuffield Department of Primary Care Health Sciences, University of Oxford, Oxford, UK
[5]Institute of Applied Health Research, University of Birmingham, Birmingham, UK

**Correspondence to**
Mr Nigel E Drury;
n.e.drury@bham.ac.uk

## ABSTRACT

**Objectives** Few children undergoing heart surgery are recruited to clinical trials and little is known about the views and attitudes of parents towards trials. This study explored parents' perspectives on decision-making about their child's participation in a clinical trial during their elective cardiac surgery.

**Design** Qualitative interview study.

**Setting** Single-centre substudy of a multicentre, double-blind, randomised controlled trial to investigate the effects of remote ischaemic preconditioning in children undergoing cardiac surgery.

**Participants** Parents of children approached to participate in the trial, both consenters and decliners.

**Methods** Semistructured interviews were conducted face-to-face or by telephone following discharge, digitally audio-recorded, transcribed and thematically analysed.

**Results** Of 46 patients approached for the trial, 24 consenting and 2 declining parents agreed to participate in an interview (21 mothers, 5 fathers). Parental decision-making about research was influenced by (1) potential risks or additional procedures; (2) personal benefit and altruism for the 'cardiac community'; (3) information, preparation, timing and approach; and (4) trust in the clinical team and collaboration with researchers. All of these were placed within the context of their understanding of the trial and knowledge of research.

**Conclusions** Parents of children undergoing cardiac surgery attach value to clinical research and are supportive of clinical trials when there is no or minimal perceived additional risk. These findings enhance our understanding of the factors that influence parents' decision-making and should be used to inform the design and conduct of future paediatric surgical trials.

**Trial registration number** ISRCTN12923441; Pre-results.

## Strengths and limitations of this study

► This qualitative substudy aimed to identify the most important issues that influence parents' decision-making on whether to allow their child to take part in a surgical trial, to improve the design and conduct of future trials.
► The interview topic guide, protocol, and study documents were developed with extensive patient and public involvement.
► The clinical trial was a suitable vehicle to explore parents' perspectives on research as the surgery was elective with low predicted mortality and the trial intervention presented minimal additional risk.
► The study reached data saturation for parents who consented to the trial but was limited by the low number of parents who agreed to be interviewed after declining their child's participation.
► Parents were recruited from a single large paediatric cardiac surgical centre in the UK which may limit generalisability.

recruited to cardiac surgical trials, all of which have been small, single-centre, phase II trials[2]; in contrast, over 70% of children diagnosed with cancer are enrolled into national or international late phase trials.[3] As a congenital heart disease community, we have a responsibility to conduct well-designed, multicentre trials to answer key questions to improve the outcomes of surgery for children and their families.[2]

Recruitment to paediatric trials is recognised to be challenging[4 5] but can be improved by understanding the factors that are important to parents when considering whether to allow their child to take part. The role of parents in this decision-making is complex, balancing the perceived risks and benefits of taking part.[6] However, little is known about the views and attitudes of the parents of children undergoing cardiac surgery towards involvement in research and specifically clinical

## INTRODUCTION

Randomised controlled trials are the accepted gold standard to evaluate the efficacy of treatments, promote evidence-based practice and improve the quality of clinical care. However, of the approximately 4500 children who undergo surgery for congenital heart disease annually in the UK,[1] less than 1% have been

trials; by understanding parents' perspectives, we can support their decision-making by improving the design and conduct of future trials. This knowledge is useful for enhancing families' experiences of trial participation and increasing recruitment,[6] thereby expanding the evidence base to guide treatment and improve patient outcomes. Qualitative studies provide participants' experiences in their own words, allowing exploration of the meanings they attribute to them, which is crucial to getting beyond assumptions about what matters in the processes of decision-making.[7] We conducted interviews with the parents of children approached to participate in a low-risk, double-blind, randomised controlled trial to explore their perspectives on research involving their child, with the aim to better understand the factors that influence their decision whether or not to participate in a clinical trial.

## MATERIALS AND METHODS

This qualitative study was a single-centre substudy of the Bilateral Remote Ischaemic Conditioning in Children (BRICC) trial, a multicentre, double-blind, randomised controlled trial of remote ischaemic preconditioning in children (ISRCTN 12923441).[8] In the trial, children aged 3 months–3 years undergoing elective surgery for either isolated ventricular septal defect closure or tetralogy of Fallot repair were recruited. Parents were provided with the trial parent/guardian information sheet (PIS) either in the clinic/ward or sent in the post and usually given at least 2 weeks, but no less than 24 hours, to consider their child's participation and ask questions. Written informed consent was obtained by a consultant, usually not the surgeon performing the operation and typically on the day before surgery. Bilateral lower limb preconditioning was performed after induction of anaesthesia but prior to sternotomy. Right atrial (additional) ±right ventricular (when routinely resected) biopsies were obtained intraoperatively and blood samples were taken from indwelling lines during the first 24 hours after surgery.

### Patient and public involvement

The substudy protocol was reviewed and amended following feedback from the Clinical Research Network's Young Person's Steering Group in the West Midlands, comprising 11 young people and 1 parent. Four parents of children who had previously undergone cardiac surgery reviewed the substudy PIS and consent forms to improve clarity and readability. Another eight parents were convened as a focus group (facilitated by NED and AL) to discuss their opinions, beliefs, concerns and expectations of research in children's heart surgery,[9] and this was used to develop an interview topic guide.

### Participants

Parents of children approached to participate in the trial, both those who consented and those who declined, were eligible to be interviewed. Potential participants were excluded if their child had experienced a serious adverse event during or immediately after surgery to avoid further distress, including death, extracorporeal life support or further surgery in the early postoperative period, or if their level of English was insufficient to participate in the interview process. Recruitment began 6 months after starting the trial to allow time for the healthcare professionals to become familiar with the trial processes. Subsequently, all eligible parents were approached to participate, and it was estimated that the parents of 20–30 children would be interviewed but recruitment would stop if data saturation was reached or trial recruitment was completed.

### Recruitment

Following discharge from paediatric intensive care unit (PICU), one or both parents were approached on the ward by a research nurse. Parents were provided with the PIS and offered the opportunity to participate in an interview at a convenient time: prior to hospital discharge, face-to-face at home, at an outpatient visit or by telephone. With written informed consent, semistructured interviews were conducted by an experienced qualitative interviewer (JCM), previously unknown to the parents and independent of the clinical team, usually within 6 weeks of discharge.

### Interviews

The schedule included: understanding of the trial, research knowledge, provision of information, timing of approach, acceptability, motivations for participation, types of research and most important factors in decision-making. As subsequent member checking by participants for accuracy and resonance with their experiences was deemed impractical, participant validation was performed during the interviews by summarising, repeating or paraphrasing the participants' words.[7] Field notes were recorded to provide context, aid interpretation, and document emotions and non-verbal behaviours. Risk to participants was deemed to be minimal but in the event of distress or concern about events which had occurred, parents were signposted to their cardiac specialist nurse, general practitioner or the Patient Advice Liaison Service, as appropriate.

### Analysis

Interviews were digitally audio-recorded, professionally transcribed, anonymised and thematically analysed[10] by three researchers (JCM, NED, AL), using NVivo V.12 (QSR International, Melbourne, Australia) for data management. The initial eight interviews were coded by all researchers independently, the coding structure compared to ensure consistency and a common coding scheme developed. The remaining interviews were allotted between the three researchers who coded these independently and convened to generate themes and discuss deviant cases.

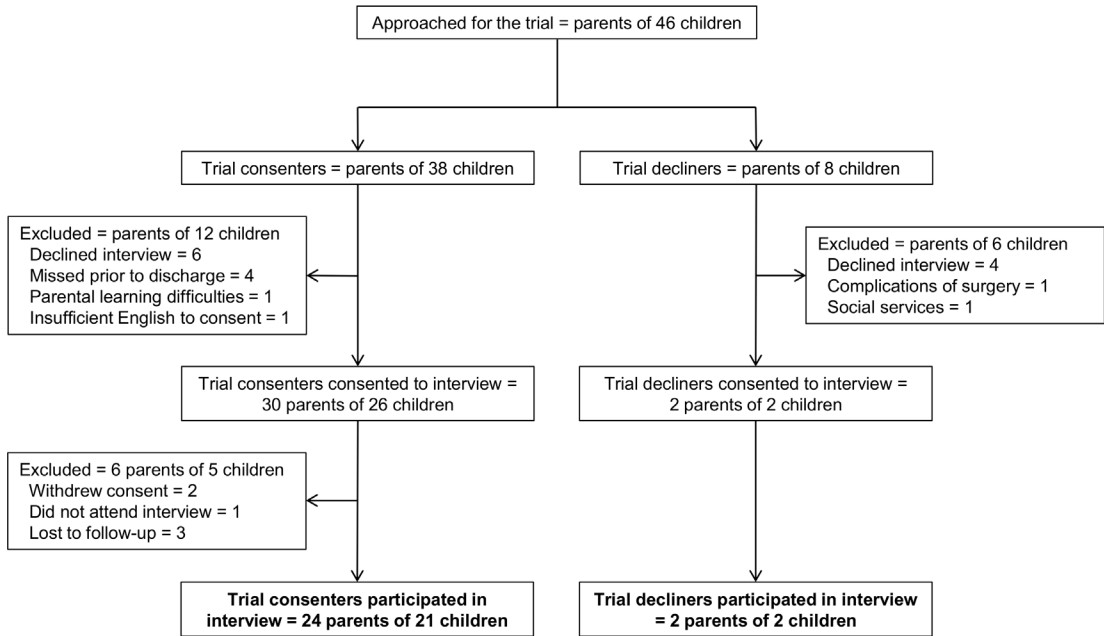

**Figure 1** Participant flow diagram.

The first authors had full access to all the data and take responsibility for its integrity and analysis.

## RESULTS

Between September 2017 and June 2019, the parents/guardians of 46 children were approached about their child's participation in the BRICC trial at the Birmingham site, of whom 38 consented to the trial and 8 declined (figure 1). Interviews were conducted with 26 parents of 23 children, 24 consenting parents (of 21 children) and 2 declining parents (of 2 children); child and participant demographics are shown in table 1, with participant-level descriptions in the online supplemental file. Seventeen (74%) interviews took place within 6 weeks of hospital discharge and all within 3 months. Data collection was stopped after interviewing 26 parents as it was agreed that data saturation had been reached for consenting parents.

Parental decision-making about whether their child should participate in cardiac surgical research was influenced by four key factors: (1) risks of participation and additional procedures; (2) personal benefit and altruism for the 'cardiac community'; (3) information, understanding and timing of approach; and (4) trust in the clinical team (figure 2). These were placed within the context of their understanding of the trial and knowledge of research. In the quotes, C indicates a consenting parent and D signifies a declining parent; additional quotes are provided in the online supplemental file.

### Risks of participation

Parental decision-making was influenced by the perceived level of risk for potential harm posed by the research. Some parents weighed up the risks and benefits of their child participating while for others, the focus on any perceived risk trumped any potential benefit; they could

| Table 1 Child and interview participant demographics | |
|---|---|
| **Children** | n=23 |
| Age at surgery, median (IQR) (months) | 9 (5–14) |
| Congenital heart disease, n (%) | |
| Tetralogy of Fallot | 12 (52) |
| Ventricular septal defect | 11 (48) |
| Hospital length of stay postop, median (IQR) (days) | 6 (5–8) |
| Siblings, n (%) | |
| None | 11 (48) |
| One | 5 (22) |
| Two or more | 7 (30) |
| **Interview participants** | n=26 |
| Relationship to child, n (%) | |
| Mother | 21 (81) |
| Father | 5 (19) |
| Age, n (%) | |
| <25 years | 4 (15) |
| 25–34 years | 13 (50) |
| ≥35 years | 9 (35) |
| Ethnicity, n (%) | |
| Caucasian | 20 (77) |
| South Asian | 4 (15) |
| Black | 1 (4) |
| Other | 1 (4) |
| Interviewees, n (%) | |
| One parent | 20 (77) |
| Both parents | 6 (23) |

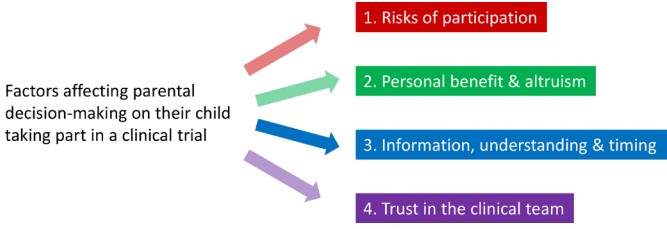

Factors affecting parental decision-making on their child taking part in a clinical trial

1. Risks of participation

2. Personal benefit & altruism

3. Information, understanding & timing

4. Trust in the clinical team

**Figure 2** Factors affecting parental decision-making.

overlook a lack of personal benefit to help others but only if there was no risk to their own child.

> I was quite open to it because there was no risk… If there'd been a risk to her, I wouldn't have done it. (C10)

The trial involved additional procedures to standard care, such as blood sampling on PICU and intraoperative tissue biopsies. Parents were largely unconcerned about the additional blood samples, even though these were 'extra', because samples were taken from indwelling lines and therefore not associated with painful procedures. If this had involved additional venepuncture, parental attitudes would have changed; for many, the added potential distress to their child would be the determining factor.

Both parents who declined the trial were apprehensive about being involved in research per se, that it is something extra and therefore has an inherent and unnecessary risk.

> They wanted to know if it would make things easier with the operation but for us, we thought they were experimenting, so we didn't want it done. Just in case something went wrong whilst they did it. I know it's just a cuff but it's just one of those things. (D2)

Although, on reflection during the interview, one parent (D1) had changed her mind and would now be willing to take part.

### Personal benefit and altruism

Parents often described considering personal benefit for their child in their decision-making. They recognised that while the trial intervention may have a direct benefit, the research may also help their child later in life, especially if they required further surgery. Several parents considered the potential benefit to any future unborn children who may also have a congenital heart defect and require surgery.

> You need to have the research for the future. I now know that there's a risk that if I have another child, there could be a heart problem. Now we've looked back on it, heart problems actually run in our family… research may help my baby in the future. (C20)

Families felt a strong sense of being part of a community affected by congenital heart disease, particularly while in hospital. Many reflected that they were benefiting from previous parents' participation in research and that they

could contribute to improving surgery and outcomes for future families. They were positive about how participating in the trial made them feel and satisfaction that their child had contributed to research. Parents who had declined the trial also recognised that participation in research may benefit other children but focused on the needs of their own child.

> Obviously, it's not like you don't care about anyone else's child but you know when you are going through it it's like you want everything the best for them. (D2)

### Information, understanding and timing

Parents described the turmoil of getting their heads around the cardiac diagnosis, what the surgery involved and how it would impact on them and their child, and it was against this background that decisions about trial participation were made. The trial PIS was written with patient and public involvement (PPI) input and most parents found the quantity and complexity of information to be appropriate.

> It was explained well, it wasn't full of, you know, big words that you don't understand and equally you know, yeah I do feel it was aimed at the right audience really. (C16)

However, there were often misconceptions about the trial, the intervention and how it related to the operation. There was a lack of confidence in, and some misunderstanding of, research terminology such as 'randomisation', 'blinding' and 'placebo'. Misconceptions involved a belief that their child was 'chosen' for the study or not understanding that the study had set inclusion and exclusion criteria.

Most parents had been sent the information by post to give them time to consider their child's involvement and discuss with family members; most found this approach agreeable and valued the time to read and reflect.

> We probably didn't read it as well as we could have, but we did screenshot the information and send it to family members and asked them their opinion, so they had time and opportunity to read it without the emotion that we were going through. (C8)

Having had the opportunity to ask questions was also important, either in the clinic or on admission to the ward. When asked about how they would feel if they had to make an urgent decision, perhaps if their child was on PICU or the surgery was imminent, there was much greater apprehension.

> I think the time issue is very important for people. If it was an emergency and their child is having surgery, they probably haven't digested that and then to receive notification about this it might be overwhelming for some people and it might have been overwhelming for us too, really. (C6)

The key reason given by both parents who declined the trial was that they felt overwhelmed by their situation, with the forthcoming operation and their responsibilities for the rest of their families; they saw the research as an additional burden on their time, energy and emotions.

### Trust in the clinical team

Families had a high level of confidence and trust in their clinical caregivers; the interviews were full of positivity about their surgeon, the wider clinical team and the National Health Service. Positive relationships between researchers and clinicians were associated with favourable perceptions of research and acceptability. Even when they did not understand the research or concepts such as randomisation, as noted above, some parents felt that their understanding, or lack of, was not important, because of their trust that the clinical team would not do anything to harm their child. On the one hand, this meant that they did not mind if they were blinded:

> It doesn't really bother me to be fair because obviously you know what you're doing, and you know what… (C18 father)

> He's in the right hands and we trust them completely. (C18 mother)

While on the other, it suggests that there is some misunderstanding about the role of research and its relationship with clinical practice. This links back to the concepts of personal risk and benefit, in suggesting that parents may not perceive the depersonalised aspects of research, such as randomisation, instead trusting that even in the context of a trial, the surgeon will 'choose' to do what is 'best' for their child.

### DISCUSSION

The diagnosis of congenital heart disease and the realisation that their child requires surgery is a particularly stressful time for parents.[11] In other conditions, the seriousness of the child's disease, risk of the intervention and urgency of participation have been identified as important influences on how parents experience recruitment, their sense of vulnerability and the success of communication[12]; however, few studies have explored parents' perspectives on clinical trials involving their child undergoing heart surgery. Hoehn *et al*[13] analysed the unsolicited comments of parents of neonates undergoing cardiac surgery in Philadelphia, Pennsylvania, USA regarding their reasons for agreeing or declining to participate in research; the most common reasons were societal benefit, individual benefit for their child and perception that it posed no harm, although parents also expressed concern about the risk of the study and anti-experimentation views. The same group evaluated parental decision-making using a competence assessment tool and found that despite the stress of surgery, parents were able to understand study-specific information and make informed decisions on their neonate's participation in research.[14] Finally,

Hoffman *et al* surveyed parents of children admitted for elective cardiac surgery in Columbus, Ohio, USA and found that 91% thought that clinical trials would improve the quality of care, while 74% believed that their child may receive direct benefit from enrolling in a trial.[15] In the present study, we explored the factors that influence parents' decision-making on their child's participation in a cardiac surgical trial and identified four key aspects relating to perceived risk, potential benefit to their child or others, information and timing of approach, and trust in the clinical team.

### Risk

As surrogate decision-makers, the over-riding consideration of parents is to act in the best interests of their child and protect them from harm.[6] Parents often feel responsible for their child's outcome in a trial and find giving consent for the child to participate much more difficult than if they were participating themselves.[4] This decision may be made more difficult by the complexity of the information and the uncertainties inherent to clinical trials, including group allocation and potential benefit from the intervention; the concepts of randomisation and clinical equipoise may be both cognitively and emotionally challenging, raising concern that their child may receive a less effective treatment.[16] The sense of responsibility for decision-making may also make parents more vulnerable to regret over making the 'wrong' decision, whether they agree to participate or not, and anticipation of regret for 'failing to protect' their child may be a major influence.[6] We found that most parents were reassured that the intervention posed no or minimal additional risk and were happy for extra blood samples to be taken if it did not negatively impact on their child. Not wanting anything extra done has been identified as a common reason for declining participation,[17] but by only obtaining blood from indwelling catheters, we avoided any additional pain or distress.

### Benefit

Most parents find research a positive experience and are motivated by feelings of 'doing something important' and 'giving something back'.[18] Numerous studies have identified altruism as an important factor in paediatric trials, independent of any potential personal benefit to their own child.[13 19–21] We found that parents felt motivated by a strong sense of belonging to a community affected by congenital heart disease who may benefit from the research. They also recognised that their child or their siblings may directly benefit from the research if they needed future surgery; parents of children with tetralogy of Fallot are counselled that there is a high chance of needing further surgery in early adult life so they may feel more invested in the advances that future research may bring.

### Information, understanding and timing

Parents are more likely to allow their child to participate if they have a greater understanding of the specific

study and broader trial concepts.[19 22] The use of PPI in refining the clarity and readability of written trial information has been shown to improve understanding[23] while educational resources, such as the Children and Clinical Studies programme,[24] can improve parental comprehension of clinical trials and be a valuable tool to aid decision-making.[25]

In the setting of elective surgery, sending out the PIS in the post provided more time for parents to consider the research and an opportunity to discuss with others, including family members and their healthcare providers.[19] Parents are more likely to take part in a trial if they feel less time pressured.[22] When asked about potentially time-critical decision-making, they were far less certain about participating, even if they were very supportive of research. Although based on projection rather than lived experience, for many, the default position would be to decline; feeling overwhelmed has been identified as the most common reason for declining participation in non-therapeutic trials on PICU[17] and parents of children undergoing cardiac surgery may be less likely to consent than other parents.[26] This suggests that recruitment involving neonates undergoing surgery in the first few days of life may be more difficult, although the 84% recruitment rate for the parents of neonates undergoing the Norwood operation in the Single Ventricle Reconstruction trial is reassuring.[27] In this group, recruitment may be optimised by prenatal trial counselling, early provision of the PIS and frequent communication.

### Trust

Parents' trust in the healthcare professionals looking after their child may influence their decision-making.[19] In this study, parents reported immense confidence in the whole multidisciplinary team and as found by others,[15] preferred for a trial to be explained by their doctor or surgeon to provide reassurance on its validity and appropriateness for their child. If this is not possible, the principal investigator or research coordinator should be introduced by the clinical team; a close and visible working relationship between researchers and clinicians may make recruitment more effective, building on the family's trust and respect for their healthcare provider[5] and reducing inappropriate or poorly timed approaches.[20] For some, that trust extended further, outweighing their need to understand the study, believing that the clinical team would only do 'what is best' for their child. It is therefore imperative that while providing information about clinical trials, the direct care team avoid explicit or inadvertent coercion which may undermine the consent process.

### Declining the trial

Previous studies have identified many reasons for parents declining consent including perceived risk, pain or distress, interference with routine care, child's clinical condition, avoiding additional medications or a placebo, parental anxiety, time pressure, inadequate information or understanding, lack of importance,

inconvenience, approached for too many studies and anti-experimentalism.[13 17 20 22] Both of our decliners referred to the trial as 'just another thing to worry about' and one expressed anti-experimentation views. Identifying ways to address these issues may improve recruitment to future trials.

### How can we improve clinical trials in children?

Our findings suggest several factors which should be considered in the design and conduct of surgical trials in children:

▶ Develop clear and accessible parent information, with input from parents of other children who have 'walked in their shoes'.

▶ Provide the information sheet well in advance of their planned surgery, for example, send by post, when feasible.

▶ Signpost to educational resources, such as the National Heart, Lung, and Blood Institute's online Children and Clinical Studies (http://www.childrenandclinicalstudies.org/), that may improve understanding.[25]

▶ Explain any potential risks associated with the trial, but separate these from the risks of the operation.

▶ Minimise additional procedures, for example, take blood only from indwelling lines.

▶ Highlight the potential benefits of the trial, without overstating any personal benefit.

▶ Build on the parents' trust in their clinical team, with a close and visible working relationship with trialists, ensuring that clinicians are well informed to discuss the trial with parents, if asked, but avoiding coercion.

### Strengths and limitations

The strengths of this study include the extensive use of PPI, with a focus group discussion to shape the topic guide for the interviews, young person input to the protocol, and parental review of the study documents to improve clarity and readability. We allowed a run-in period for the trial so the impact of any familiarisation phase would be minimised. Most parents who were approached for interview were happy to take part, enabling us to reach data saturation for consenting parents in a timely manner, within the duration of trial recruitment. Interviews were arranged at the convenience of parents, either in person at home, in the clinic or by telephone, to facilitate their participation and were conducted by a senior nurse researcher, skilled in conducting qualitative interviews but independent of their clinical team to reduce the risk of confirmation bias or a halo effect. The BRICC trial was a suitable vehicle to explore parents' perspectives on clinical research as the intervention (remote ischaemic preconditioning) presents minimal risk, the surgery is performed electively and the operations included have a low predicted mortality (STAT categories 1–2).[28]

The limitations include only two interviews with parents who declined the trial, providing limited insight into the actual reasons for declining. This was a consequence of both the high overall consent rate in the trial,

approximately 85%, limiting the pool of decliners, and that most parents who declined the trial also declined to be interviewed; these parents are a seldom heard group who may have seen the approach for interview as intrusive or seeming to question their decision not to take part. As interviews were conducted following a period of intense stress and may have taken place up to 3 months following hospital discharge, there was potential for recall bias relating to their earlier thoughts and decision-making.[29] The study was also limited to a single, high-volume paediatric cardiac surgical centre in the UK which may limit generalisability.

## CONCLUSIONS

Parents of children undergoing cardiac surgery attach value to clinical research and are supportive of clinical trials. The most important factors that influence decision-making on whether to allow their child to take part are perceived risk, potential benefit either to their child or others, information and timing of approach, and trust in their clinical team. Trial recruitment and retention may be improved by addressing communication and information needs, particularly surrounding potential risk, and improving collaborative working with clinicians. Our findings contribute to knowledge surrounding the acceptability of research in children undergoing surgery and should be used to inform the design and conduct of future clinical trials.

**Acknowledgements** We thank Dr Jonathan Ives, University of Bristol, for his advice on development of the protocol and Helen Winmill, Jenna Spry and colleagues in the PICU Research nursing team for assistance in recruitment. We are grateful to the Young Person's Steering Group for advice on the protocol, the trustees of Young at Heart for feedback on the study documents, and Sandra Ramsey and Martina Ponsonby for help with the focus group. We thank our surgical colleagues, Ms Natasha Khan, Mr Phil Botha and formerly Professor David Barron, for their contribution to the BRICC trial, and The Transcription Company for transcription of the digital recordings. We are most grateful to the parents who either participated in the focus group or were interviewed for this study.

**Contributors** NED—conceptualisation, data curation, formal analysis, funding acquisition, methodology, resources, writing the original draft and review, and editing. JCM—data curation, formal analysis, investigation, methodology, project administration, writing the original draft and review, and editing. CJT— conceptualisation, methodology, writing review and editing. TJJ—conceptualisation, funding acquisition, writing review and editing. ACL—data curation, formal analysis, methodology, supervision, writing review and editing.

**Funding** This work was supported by a grant from the Birmingham Children's Hospital Research Foundation (BCHRF442) and an Intermediate Clinical Research Fellowship from the British Heart Foundation (FS/15/49/31612) awarded to NED. Lay review of study documents was funded by a PPI Bursary (RDS/WM-1318) from the National Institute for Health Research (NIHR) West Midlands Research Design Service. JCM is an NIHR 70@70 Senior Nurse and Midwife Research Leader, and CJT is funded by an NIHR Academic Clinical Lectureship.

**Disclaimer** The views expressed are those of the authors and not necessarily those of the NHS, Department of Health, NIHR, Birmingham Women's and Children's NHS Foundation Trust, British Heart Foundation, University of Birmingham, or University of Oxford. Neither the sponsor nor funders had any role in the design of this study, data collection, interpretation or decision to submit the results for publication.

**Competing interests** CJT reports personal fees from Novartis and Vifor, outside the submitted work.

**Patient consent for publication** Not required.

**Ethics approval** Both the clinical trial (16/WM/0309, 5 August 2016) and this qualitative substudy (17/WM/088, 28 April 2017) were approved by the West Midlands-Solihull NHS Research Ethics Committee and sponsored by the University of Birmingham (RG_14-025 and RG_17-005, respectively).

**Provenance and peer review** Not commissioned; externally peer reviewed.

**Data availability statement** The original data are available from the corresponding author upon reasonable request.

**ORCID iDs**
Nigel E Drury http://orcid.org/0000-0001-9012-6683
Julie C Menzies http://orcid.org/0000-0003-2080-3364
Clare J Taylor http://orcid.org/0000-0001-8926-2581
Timothy J Jones http://orcid.org/0000-0001-8870-3496
Anna C Lavis http://orcid.org/0000-0002-1080-2512

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
