## [Reviewer comments · BMJ Open]

ARTICLE DETAILS

TITLE (PROVISIONAL)	Understanding parents' decision-making on participation in clinical trials in children's heart surgery: a qualitative study
AUTHORS	Drury, Nigel; Menzies, Julie; Taylor, Clare; Jones, Timothy; Lavis, Anna

VERSION 1 – REVIEW

REVIEWER	K Sarah Hoehn University of Chicago Comer Children's Hospital Chicago IL USA
REVIEW RETURNED	26-Oct-2020

GENERAL COMMENTS	This is a very well written study interviewing parents of children who had cardiac surgery and were invited to participate in research studies. Their qualitative methodology with independent reviewers generating themes is excellent. Their results and conclusions give the field specific advice (advance notice, simpler language) for ways to improve this population of children participating in research trials. Their analogy with pediatric oncology trials is thought provoking and demonstrates why this type of research is so important.
--

REVIEWER	Katharine O'Hearn Children's Hospital of Eastern Ontario Children's Hospital, Ottawa, Ontario, Canada
REVIEW RETURNED	28-Oct-2020

GENERAL COMMENTS	This is a very interesting and well-written manuscript exploring an important topic in pediatric congenital heart disease. The extensive use of patient and family input into the design of both the trial and the current sub-study is an important strength. I have only minor comments for the authors to address: Materials and Methods First paragraph: In order to better understand these results in the context of the BRICC trial, the reader would benefit from additional information regarding the consent process used for BRICC. For example, when (how far from the surgery date) was the PIS typically mailed to families? Who had the follow-up conversation with the parents (research team, clinical team) and when (on the day of surgery, at a pre-operative appointment). Results Page 8, Line 19: Please clarify that data saturation for consenting parents only was reached. With only two parents of families who declined consent participating, it is difficult to accept that data
--

	saturation was reached for this group. Discussion Page 16, lines 11-16: Based on the comments from some parents that trust in the care team outweighed their need to understand the study, is there any concern that the recommendation for the care team to be involved in consent discussion could lead to coercion or provision of parental consent that is not truly informed? Please address this potential concern in the discussion. Limitations Although mentioned in the abstract, the main manuscript does not acknowledge the potential limitations of generalizability of the results (single centre in the UK). In addition, recall bias is another potential limitation, as interviews were conducted in the month following (hospital?) discharge. Depending on the timing of the initial consent discussion, and length of PICU/hospital stay, a significant amount of time could have passed between the consent decision for BRICC and the sub-study interview. In stressful situations (PICU) parents do not always remember important consent information (Gertsman et al., 2020) – given that the parents in this study were likely under significant stress prior to surgery at the time of the BRICC consent discussion, this could have further impacted recall bias. General Parents whose child experienced an adverse event during or after surgery were excluded from the current sub-study. Was there any difference for this group and the children that were eligible for the sub-study (e.g. did the children who experience an intraoperative or post-operative adverse event have a more significant lesion, or other chronic health conditions that pre-disposed them to a poorer outcome). If so, is there any concern that the viewpoints of this group would be different and should this be acknowledged as a limitation?
--	--

VERSION 1 – AUTHOR RESPONSE

We are most grateful to the reviewers for their kind and thoughtful comments. In response to the specific points raised by Reviewer 2:

2] Materials and Methods, first paragraph: In order to better understand these results in the context of the BRICC trial, the reader would benefit from additional information regarding the consent process used for BRICC. For example, when (how far from the surgery date) was the PIS typically mailed to families? Who had the follow-up conversation with the parents (research team, clinical team) and when (on the day of surgery, at a pre-operative appointment).

Response: To clarify the recruitment process, we have added two sentences to the first paragraph describing the timing of the approach, and when/by whom the follow-up conversation and consent took place (page 5).

3] Results, Page 8, Line 19: Please clarify that data saturation for consenting parents only was reached. With only two parents of families who declined consent participating, it is difficult to accept that data saturation was reached for this group.

Response: Yes, data saturation had been reached only for consenting parents and we have clarified this by adding 'for consenting parents' to the sentence (page 8).

4] Page 16, lines 11-16: Based on the comments from some parents that trust in the care team outweighed their need to understand the study, is there any concern that the recommendation for the care team to be involved in consent discussion could lead to coercion or provision of parental consent that is not truly informed? Please address this potential concern in the discussion.

Response: This is an important point which we had not acknowledged. We therefore have added a sentence to the discussion (page 16): 'It is therefore imperative that whilst providing information about clinical trials, the direct care team avoid explicit or inadvertent coercion which may undermine the consent process.' We have also added a clause about avoiding coercion to the final bullet point on improving surgical trials in children (page 17).

5] Although mentioned in the abstract, the main manuscript does not acknowledge the potential limitations of generalizability of the results (single centre in the UK).

Response: We have added a sentence on generalisability to the limitations section of the discussion (page 18), similar to the final bullet point in the strengths and limitations part of the abstract: 'The study was also limited to a single, high-volume paediatric cardiac surgical centre in the United Kingdom which may limit generalisability.'

6] Recall bias is another potential limitation, as interviews were conducted in the month following (hospital?) discharge. Depending on the timing of the initial consent discussion, and length of PICU/hospital stay, a significant amount of time could have passed between the consent decision for BRICC and the sub-study interview. In stressful situations (PICU) parents do not always remember important consent information (Gertsman et al., 2020) – given that the parents in this study were likely under significant stress prior to surgery at the time of the BRICC consent discussion, this could have further impacted recall bias.

Response: We have added a sentence on recall bias to the limitations (page 18) including the suggested reference: 'As interviews were conducted following a period of intense stress and may have taken place up to three months following hospital discharge, there was potential for recall bias relating to their earlier thoughts and decision-making [29].'

We have also clarified in the methods (page 6) and results (page 8) sections that interviews usually took place within 6 weeks but some occurred after this time due to logistical constraints of arranging a suitable time for interview with the parent(s). In the first version of the manuscript, our statement of within one month was taken from our original protocol and was included in error; the period was extended to within three months of discharge as the challenges of making arrangements with the parents emerged during the study. In the methods section, we have also clarified that, for the timing of interviews, discharge relates to hospital discharge (page 6).

7] Parents whose child experienced an adverse event during or after surgery were excluded from the current sub-study. Was there any difference for this group and the children that were eligible for the sub-study (e.g. did the children who experience an intraoperative or post-operative adverse event have a more significant lesion, or other chronic health conditions that pre-disposed them to a poorer outcome). If so, is there any concern that the viewpoints of this group would be different, and should this be acknowledged as a limitation?

Response: The exclusion of parents whose child had experienced a serious adverse event was used to avoid undue distress. As shown in the flow diagram (figure 1), only one patient was excluded due

to a postoperative complication, requiring further surgery in the early post-operative period. We therefore do not believe that this was a limitation of the study.

VERSION 2 – REVIEW

REVIEWER	Katharine O'Hearn Children's Hospital of Eastern Ontario, Canada
REVIEW RETURNED	01-Dec-2020
GENERAL COMMENTS	The authors have addressed all of my previous comments. I do not have any additional comments at this time.